# Positive Orientation as a Predictor of Health Behavior during Chronic Diseases

**DOI:** 10.3390/ijerph16183408

**Published:** 2019-09-14

**Authors:** Ewa Kupcewicz, Anna Szypulska, Anna Doboszyńska

**Affiliations:** 1Department of Nursing, Faculty of Health Sciences, Collegium Medicum University of Warmia and Mazury, 10-719 Olsztyn, Poland; 2Pulmonology Department, Faculty of Health Sciences, Collegium Medicum University of Warmia and Mazury, 10-719 Olsztyn, Poland

**Keywords:** positive orientation, health behavior, chronic diseases, health

## Abstract

*Background:* Positive orientation includes optimism, self-esteem, and life satisfaction. This research was conducted to determine whether positive orientation is an important predictor of health behaviors of patients with chronic movement disorders which require a rehabilitation program. *Methods:* The study involved 93 patients, including 46 women (49.5%) and 47 men (50.5%). The study utilized a standardized Positive Orientation Scale and a Health Behavior Inventory to measure the intensity of health-promoting behaviors. *Results:* The two variables of age and positive orientation were found to be predictors of overall healthy behavior, explaining a total of 22% variance of the dependent variable. Age was a predictor of preventative behavior (β = 0.37; R^2^ = 0.16). Its share in the prediction of this variable was significant (16%). The highest predictive value of positive mental attitude was having a positive orientation (β = 0.42; R^2^ = 0.17), which explained 17% of the variance of the dependent variable. However, age explained 14% of the variability of health practice results (β = 0.39; R^2^ = 0.14). *Conclusion:* The results of the research indicate the need to implement preventive programs with a positive orientation to modify the health behavior of chronically ill male and female patients.

## 1. Introduction

Several terms and classifications of healthy behaviors are specified in the research literature. Steptoe et al. have identified five classes of healthy behaviors which include dietary habits, substance avoidance, communication safety, positive health practices, and preventive behavior [1]. The results of many scientific studies point to the link between healthy behaviors and the state of health [2,3]. Hamer et al. showed the relationship between physical activity and the mental state. They found that even light exercise significantly reduced stress and anxiety [4]. Other studies indicate that people who regularly exercise have a greater sense of optimism and excitement [5]. It has been shown that physical activity has a positive effect on cognitive performance [2]. Moderate physical activity also prevents cardiovascular disease, reduces serum cholesterol concentration, and stabilizes lipid metabolism [6,7]. On the other hand, a scarcity of physical activity may lead to serious health consequences. Cooper et al. showed the negative effect of a sedentary lifestyle, which may be a direct or an indirect cause of death [8]. Worth noting is the study by Inoue et al. of a group of 83,034 Japanese citizens aged 45–74. The study analyzed the association between daily total physical activity and the risk of all-cause mortality and morbidity from cancer, heart disease, and cerebrovascular disease. During follow-ups, there were 4564 deaths. The results indicated that a decreased risk was observed regardless of age, frequency of leisure-time sports, physical exercise, or obesity status, albeit with an important risk attenuation among those with a high body mass index. A significant, decreased risk was also observed for death from cancer and heart disease in both men and women, as well as from cerebrovascular disease in women [9]. Health behaviors are becoming increasingly important issues for researchers who believe that the personal resources of individuals are important in shaping healthy behaviors [10]. The results of many studies have shown a relationship between healthy behavior and a sense of self-efficacy, a sense of coherence, an internal locus of health control, and a sense of optimism [3,5,10,11]. In recent years, there has been dynamic development in the field of positive psychology, which has resulted in a number of concepts becoming a good foundation for research on the most adaptable forms of psychosocial functioning [12]. One of the concepts developed within this trend is the theory of positive orientation, which was first formulated by Gian Vittorio Caprara, an Italian psychologist at the University of La Sapienza in Rome. The first work from Caprara appeared in 2009, and coined the term “positive orientation” [13]. The results of positive orientation studies indicate that this is a “fundamental tendency to notice and to attach importance to the positive aspects of life, experience and self. It is largely responsible for adaptive functioning, which means a natural inclination towards positive self-esteem, high satisfaction in life and high evaluation of opportunities for achievement, which translates into commitment to life’s aspirations and a high quality of life” [14,15,16]. The results of empirical studies have shown that positive orientation combines three components: optimism, self-esteem, and satisfaction from life [15,17]. Italian, Canadian, German, and Japanese studies confirm that a positive orientation can be considered to be a “good functioning syndrome”, which correlates positively with the state of health [18]. Other studies have indicated a relationship between positive orientation and belief in self-efficacy [18]. An a posteriori study presented by Caprara et al. indicates that a positive orientation, seen as a relationship of self-efficacy and positive perception of one’s self-esteem and one’s past and future experiences, which promotes a higher level of commitment to life, can have a positive impact, not only on an individual but also on a social group, and constitutes a significant personal resource which is important in the context of workplace environment [19,20]. Furthermore, Caprara et al. hypothesized that its positive influence can constitute a major intermediary psychological mechanism in behavioral expression of positive orientation [21]. Laguna was one of those who undertook to verify the hypothesis and found “the beliefs-affect-engagement model, postulating that positive orientation stimulates positive affect which, in turn, fosters activity engagement”, and can be applied to many other spheres of human life [22]. On the other hand, Oleś and Jankowski investigated the issue scientifically and presented a broader understanding of the term positive orientation “which is conceptualized, here, as a latent factor underlying variables that exemplify a hedonistic and eudemonistic view of happiness” [23]. Tisak made the assertion that a positive orientation had an impact on perceived social support as well as on building friendly social relationships. Rich resources of positive orientation can be activated in situations of higher expectations, stress, or health problems [19]. As Caprara et al. claim, positive orientation constitutes a health factor which supports psycho-physical wellbeing [20]. Illness is perceived by an individual as a new situation. Having experienced the first symptoms or having been diagnosed, a patient begins the process of searching, selecting, and implementing activities to adapt to new conditions required by illness. A patient’s activity during illness is reflected in the patient’s manner of coping with stress and it is related to the process of treatment and rehabilitation or it also involves the actions self-initiated by a patient. One type of such activities involves healthy behaviors during illness. They comprise both abandoning unhealthy behaviors as well as introducing health-promoting practices in one’s lifestyle. The authors of the paper claim that a patient’s personal resources may trigger their willingness to change toward healthy behaviors, provided such a change is necessary or desired for health improvement. However, in recent years, the dynamic development of research has focused on identifying new psychological components of personal resources, with positive orientation being one of them, and as shown in the research review, it is still required to identify the factors which trigger a patient’s willingness to fight and motivation to undertake activities in the area of healthy behaviors. The authors of the paper also draw attention to patients’ problems resulting from exhaustion or lack of personal resources. According to the research review, an occurring problem in the area of public health involves the syndrome of frailty, which on an individual level, encompasses the losses in the following domains: physical, psychological, and social [24]. Sacha et al. note that people with frailty possess reduced potential to cope with external stressors and to respond to life incidents. As a result, such individuals are susceptible to adverse consequences such as: falls, impairment of cognitive function, infections, hospitalization, disability, institutionalization, and death [25].

Increasing interest in prevention programs and behavior modification has intensified the search for predictors of health behavior, among which a positive orientation may find a significant place.

The following research questions were formulated for further investigation in the paper:What is the role of positive orientation in the prediction of intensified general health-promoting behaviors and in the intensified occurrence of four categories of healthy behaviors (i.e., proper eating habits, preventive practices, healthy practices, and positive mental attitude) in the study group subjected to the rehabilitation program?Is there diversification in the intensity of general health-promoting behaviors beneficial for both men and women subjected to a rehabilitation program, and if so, to what extent?

The following research hypotheses were tested:Positive orientation, as a personal resource, explains, to a significant degree, intensification of healthy practices related to a positive mental attitude, and it determines, to a less significant extent, intensification of general health-promoting behaviors as well as healthy behaviors related to proper eating habits, preventive practices, and healthy behaviors.Both men and women show an equal level of intensification of health-promoting behaviors.

## 2. Material and Methods

### 2.1. Sample Characteristics

A total of 104 patients, hospitalized in the Clinical Department of Neurological and Day Care Rehabilitation Ward of the University Clinic in Olsztyn (Poland) between March, 2014 and February, 2015, were invited to participate in the research. While aggregating the data, 11 questionnaires were found to be incomplete. Following this, 93 patients, undergoing treatment for 6–16 weeks continuously in the hospital environment, were included in the study group. The research involved 94.6% (n = 88) orthopedic patients subjected to orthopedic rehabilitation related to organ movement injuries, degenerative diseases, or acquired osteoarticular and muscular system changes, and 5.4% (n = 5) stroke patients with general motor disorders. The poststroke disability types involved a difficulty or inability to make an active deliberate movement and problems with keeping balance and keeping a standing position of the body. The mean age of participants was 58.9 (±14.13) and the median age was 61.0 years. The largest groups were patients aged 61–70 years (n = 32; 34.4%), living mostly in the city (n = 66; 71.0%). Only one-quarter of the respondents were active and 39.8% (n = 37) were retired. Most of them were people with secondary education (n = 39; 41.9%). Half of the respondents declared that they had some degree of disability (n = 47; 50.5%). A high percentage of subjects (n = 66; 71.0%) had a BMI (kg/m^2^) above normal (<30). For over half of the patients participating in the research (n = 48; 51.6%), the period of planned rehabilitation took six weeks, and for the remaining group (n = 45; 48.4%), the period was prolonged to 16 weeks. Up to 41.9% (n = 39) of the respondents suffered from systemic illnesses, the most frequently indicated of which was hypertension.

### 2.2. Procedure

The study was conducted in line with the principles outlined in the Helsinki Declaration. Permission to carry out the study was obtained from the Research Ethics Committee at the University of Warmia and Mazury in Olsztyn (Ordinance 19/2012). Subject sampling was justified. The main criteria for inclusion in the study group involved: the diagnosis of a chronic illness (as defined by the National Commission on Chronic Illness) requiring special rehabilitation, surveillance, observation, or care [26] as well as a period of rehabilitation including at least six continuous weeks, patients belonging to a homogenous diagnosis-related group (JPG), and the patient’s written consent to participate in the study. The major criterion of exclusion from the research concerned patients with disturbed perception of the external world. Prior to the study, the patients were informed about the aim of the survey, were encouraged to ask questions, and were provided with full explanations. The patients had the right to withdraw from the research at any time and without specifying their reasons. After providing written consent to voluntarily participate in the research, the participants were provided a file of questionnaires to be completed, which were then submitted to one of the researchers within two days. All of the patients received one questionnaire designed by the author of the paper referred to as the Own Design Questionnaire and two standardized research tools: the Positive Scale Guideline [18] and the Health Care Inventory [10]. The participants who had difficulty in writing were assisted by the researchers. It took 20 minutes to complete the questionnaires. 

### 2.3. Own Design Questionnaire

The sociodemographic variables (i.e., gender, age, place of residence, marital status, education, financial situation, occupational activity, rehabilitation period, systemic illnesses, and the degree of disability) were identified using the self-assessment questionnaire. A detailed clinical history was compiled and body mass index (BMI) was calculated.

### 2.4. The Polish Adaptation of the Positive Orientation Scale

The level of positive orientation was diagnosed using the Positive Scale Guideline by G. V. Caprara et al. [18]. The Polish adaptation was done by M. Łaguna, P. Oles, and D. Filipiuk [18]. The scale is a self-reporting tool that contains eight statements, all of which have a diagnostic character. The subjects were asked to indicate to what extent they agreed with each statement. Answers were given on a five-point scale from 1 (I strongly disagree) to 5 (I strongly agree). The sum of all points was a general measure of the level of positive orientation and ranged from 8 to 40 points. Higher scores were correlated with higher levels of positive orientation. Sten standards were developed for standardized tests. The psychometric properties of the Polish version of the scale were satisfactory and the values of the Cronbach α coefficient were in the range of 0.77 to 0.84 [18].

### 2.5. The Polish Adaptation of Healthy Behavior Measures

The Health Care Inventory (IZZ) by Z. Juczynski [10] contains 24 statements describing various health behaviors and has been used to measure healthy behaviors. By analyzing the frequency of individual behaviors indicated by the respondents, the overall health-promoting behaviors and the severity of the four categories (subscales) of health behaviors were calculated.

The indicated subcategories of healthy behavior were:proper eating habits (taking into account the type of food preferably consumed);preventive behaviors (related to compliance with health recommendations, information on health, and illness);health practices (including daily habits of sleep and recreation or physical activity);positive mental attitude (related to avoiding intense emotions, stresses, and tensions) [10].

Respondents indicated how often they performed health-related activities in the past year, evaluating each one of them on a five-point scale. Answer scores were as follows: 1—almost never, 2—rarely, 3—occasionally, 4—often, 5—almost always. The overall scale score was the sum of all points with a distribution between 24 and 120 points. Higher scores were correlated with a greater intensity of declared healthy behaviors. The overall indicator in the standardization study was transformed into standardized units and interpreted on a stenographic scale. The intensity of the four categories of healthy behaviors was calculated as the average number of points in each category divided by six. The psychometric properties of the scale were considered to be satisfactory, the Cronbach α coefficient was 0.85 for the entire Inventory, and for its four subscales, it ranged from 0.60 to 0.65. The stability index measured at six-week intervals by a retest was 0.88 [10].

### 2.6. Statistical Analysis

The data generated during a posteriori study were subjected to statistical analysis using the Polish version of STATISTICA 13 (TIBCO, Palo Alto, CA, USA). For the statistical analyses, the following were used: the measures of location and variability, t-test for independent samples, and the r- Pearson correlation parameter. A nonparametric Kolmogorov–Smirnov test was applied to determine whether the variables featured normal distribution. The overall rate of positive orientation and healthy behaviors was translated into standardized units with regard to the values of a standard ten scale. A Sten score contains 10 units where one deviation equals one sten. The results within 1–4 stens were regarded as statistically weak, 5–6 stens as statistically average, and 7–10 as statistically significant [10,18]. In order to analyze the distribution of research results for the study groups of men and women according to the standard ten scale, the chi-square test (χ^2^) was applied. The research hypothesis was tested using multiple regression modeling for estimating quantitative relations among several independent variables (predictors) and a dependent variable (the criterion variable). The relationship between the independent and dependent variables was expressed as R, defined as the square root of R-squared. The values of R were within the range of [0,1]. The regression parameter (β) was used to specify the direction of dependence. The statistical significance value applied was *p* < 0.05.

## 3. Results

The mean score for overall healthy behaviors in the study group was 91.39 (±17.56) points, while the mean for the four subscales of healthy behaviors ranged from 3.67 to 3.99, taking the highest point value for the health behavior associated with positive mental attitudes (Table 1). The mean for the positive orientation was 29.14 (±5.11).

Subsequently, the raw data were translated into standardized units with regard to the values of a standard ten scale for both the positive orientation [18] and general severity of healthy behaviors [10] in order to compare the obtained values. More than a quarter of the respondents (28.0%) received high results of 7–10 stens, showing a high level of positive orientation. This indicates that the respondents were optimistic and were inclined to attach importance to the positive aspects of life and self [5]. Nearly half of the respondents (45.2%) achieved results in the range of 5–6 stens, indicating an average level of positive orientation. Results between 1 and 4 stens were collected from 26.9% of respondents. 

The subsequent analyses showed that the patients most frequently represented a high level of health-promoting behaviors. High results were obtained by 60.2% of respondents, whereas average scores were obtained by 23.7%, and low scores only by 16.1%. The distribution of results was similar for both men and women (χ^2^ = 0.66; *p* < 0.71). Figure 1 shows the distribution according to Sten scores for general severity of healthy behaviors and positive orientation in the study group.

In the research, the relationship between the level of positive orientation and healthy behaviors was investigated taking into account the gender of the respondents. Statistical analyses using the r-Pearson correlation coefficient and the relationship between the variables were based on Guilford’s classification. The results show that there is a statistically weak, but statistically significant, positive correlation between positive orientation and general healthy behaviors (r = 0.29; *p* < 0.01), appropriate eating habits (r = 0.24; *p* < 0.01), and healthy practices (r = 0.25; *p* < 0.05). There is also a significant positive correlation with positive mental attitude (r = 0.41; *p* < 0.001). This means that higher levels of positive orientation are correlated with the respondents showing positive healthy behaviors, both generally and in proper eating habits, health practices, and positive mental attitudes (Table 2). In addition, positive orientation showed a positive correlation with positive mental attitude in groups of women (r = 0.34; *p* < 0.01) and men (r = 0.45; *p* < 0.01). It may be concluded that the higher the level of positive orientation, the more often respondents show a positive mental attitude, which is consistent with the concept of positive orientation theory [16].

In the course of further study, progressive multiple regression modeling was applied in order to test the research hypotheses. During the initial stage of building the regression model (stage 0), the independent variables (predictors) were selected from the group of 12 preliminary variables (gender, age, place of residence, marital status, education, financial situation, disability statement, occupational activity, BMI, hospitalization period, systemic illnesses, and positive orientation). Eventually, as a result of statistical analysis, two variables were selected for the subgroup of independent variables: age and positive orientation, which were tested in stages 1 and 2 of the multiple regression model of the analysis. The other independent variables were outside of the regression model. The regression parameters indicated the contribution of the remaining independent variables in the final model to providing an explanation of the variables characteristic of the healthy behaviors of an individual. The regression model including step 1 and 2 for the predictors of healthy behaviors is presented in Table 3.

The data presented in Table 3 show a positive relationship between the variables subjected to analysis, both variables related to the severity of healthy behaviors as well as variables related to the four categories (subscales) being scrutinized. Age and positive orientation proved to be the predictors of the severity of general health-promoting behaviors explaining overall 22% (R^2^ = 0.22; β = 0.31) of the dependent variable. During stage 1 of the regression model, age explained 13% of the dependent variable, whereas during stage 2, positive orientation explained 9% of the criterion variable. Healthy behaviors referred to as proper eating habits were positioned at the same level by age during stage 1 of the regression model and by positive orientation during stage 2, scoring the overall predictive power of 12% (R^2^ = 0.12; β = 0.25). However, as for the predictor of preventive behaviors in compliance with the health recommendations, age was found to constitute the only predictor. The age variable entered the model during stage 1 and its relation to the prediction of preventive behaviors scored 16% (R^2^ = 0.16; β = 0.40). Positive orientation played an important role in the prediction of severity of healthy behaviors linked to positive mental attitude, which patients could demonstrate by avoiding intense emotions, stress, and tensions, or other depressing situations. It explained 17% of the dependent variable. The overall predictive power of positive orientation and age for the above category of healthy behaviors was 22% (R^2^ = 0.22; β = 0.42). As for the final category of healthy behaviors referred to as health practices including daily habits of sleep and recreation, the variables of age and positive orientation altogether explained 21% (R^2^ = 0.21; β = 0.26) of the data variability. Age showed the greatest contribution during stage 1, explaining 14%, whereas positive orientation in stage 2 explained 7% of the variability of the dependent variable. The statistical analyses proved that the research hypotheses were verified to a positive effect. The two variables identified (positive orientation and age) as the predictors of general severity of healthy behaviors and the correlation curve are presented in Figure 2.

## 4. Discussion

The conducted research is in line with other studies seeking to discover the psychological determinants of healthy behaviors. The study found that positive orientation was verified as a predictor of the dependent variable, defined as general healthy behaviors, and four categories of behaviors, including proper eating habits, preventive behaviors, healthy practices, and mental attitude. The variables analyzed within the course of the research prove to be related. Positive orientation and age play a significant role in undertaking and maintaining health-promoting activities, during times of both health and times of illness. However, their predictive power varies for particular healthy behaviors. The mean for positive orientation in the study group reached 29.14 (±5.11) points and was at approximately the same level as the findings presented by other researchers [10,27]. The general parameter of healthy behavior had the value of 91.39 (±17.56) points, which was slightly higher compared to the normalized research results concerning adults (81.82 (±14.16)), patients undergoing dialysis (83.45 (±14.76)), men after heart attack/myocardial infarction (84.00 (±16.54)), and women during menopause (85.98 (±12.70)), as well as women during pathological pregnancy (90.18 (±12.78)) and patients with diabetes (92.44 (±11.59)) [10].

### 4.1. Proper Eating Habits

Eating regime constitutes a crucial issue for chronically ill patients and elderly patients. The category of healthy behaviors referred to as proper eating habits scored the lowest in the statistical analyses (3.67 ± 0.91). This is likely due to the fact that healthy behaviors connected with the type and frequency of consumed food constitute a decisive factor in the irrational manner of food consumption within the study group. The authors’ own research identified higher mean values of BMI (kg/m^2^), which reflected the body mass in 71% of the patients as being above average. Szatkowska et al. found that 69% of the patients during hospitalization resulting from heart attack [28] had excessive body mass. De la Torre et al., after 10 years of observations concerning patients with lung transplants, indicated that there was a link between either overweight or obesity and more frequent cases of diabetes, hypertension, and dyslipidemia [29]. Moreover, Guida et al. conducted research in two groups of patients, one group complying with the diet and the other not following the diet. The authors concluded that a proper diet contributed to a decline of cholesterol and triglyceride levels, stabilized sugar content in blood, and supported body mass reduction, and contributed to a reduction of the adipose tissue of the patients after heart transplantation [30]. Other studies have suggested that proper diet in rheumatoid arthritis is likely to affect its severity [31,32]. However, there are also reports which do not support this relationship [31,33]. As the research review indicates, both lifestyle and preferred healthy behaviors contribute to chronic lower limb ischemia, causing its progress [34,35]. According to the forecast for subsequent years (until 2035), the Polish population is likely to develop obesity and diabetes significantly more frequently [28,36,37,38]. Studies by other authors have also indicated that proper eating habits, compared to other healthy behaviors, remain at the lowest level, especially among men and the elderly [28,39,40,41]. In the discussion above, the authors of the paper present only part of the researched topic. In order to acquire deeper knowledge of the issue, it is worth referring to a number of researchers investigating frailty, a factor related to proper eating habits [42,43]. The criteria used for recognition of frailty include criteria related (to a higher or lower degree) to unhealthy eating habits. They involve the following: unmeasured body weight loss, low muscle power, the feeling of exhaustion, lower physical activity, and lower pace of movement [42,43]. The research carried out by Fried et al. showed an independent path of development of the weakness syndrome associated with the weakening of physiological reserves leading to a significant weakening of the organism, while not detecting comorbidities or disabilities [44]. In addition, Koch et al. showed that the diagnosis of weakness syndrome increases the possibility of individually selected and targeted treatment and rehabilitation programs [45].

The treatment of frailty requires a holistic approach and complex long-term care provided for patients. The authors’ own research results confirm the correlation between positive orientation and proper eating habits, which can certainly be applied to a similar group of patients.

### 4.2. Positive Mental Attitude

In the authors’ own research, positive orientation was observed to possess a significant predictive power with reference to a positive mental attitude, which means that it can constitute a health-promoting factor and can contribute higher resistance to stressful factors and can lead to escalation of positive emotions, which constitute the source of human energy. Several studies have shown that the components of positive orientation, including optimism, self- assessment, and life satisfaction, are related to healthy behaviors. Lipowski found that optimism was correlated with proper eating habits, physical activity, and exercise as well as with sleeping habits and relaxation [5]. Steptoe et al. claimed that dispositional optimism was linked with the lifestyle of the residents of a social healthcare home. The health behaviors which were more frequently manifested by elderly optimists included lack of cigarette addiction, moderate alcohol consumption, and regular physical exercise [46,47]. Likewise, Giltay et al. argued compellingly in favor of such a point of view [46,48]. They asserted that a low level of optimism can be directly linked to cardiovascular illnesses and can increase the risk of death due to unhealthy behaviors [41,46]. As the research suggests, social relationships also contribute to the better health condition of an individual. During the hospitalization of a patient, a significant role is attributed to the therapeutic communication [49]. Bunt et al. also noted the adverse condition of an individual, referred to as social frailty, which is related to the risk of losing or actual loss of resources important for satisfying fundamental social needs during one’s lifetime [50]. Social support, which constitutes a personal psychological resource of an individual, contributes significantly to life satisfaction, quality of life during illness, and alleviating the negative effects of stressful events [51]. Juczyński presented the importance of social support for the well-being of an individual using the research conducted with a Social Support Questionnaire (Soz-U K-14) [51]. According to Juczyński, the lowest social support was revealed by a group of elderly adults, regardless of their gender; additionally, the social support significantly correlated with age. People under 60 manifested higher social support than older individuals [52]. However, Kojima et al. noted a quality of life problem of the people with frailty syndrome, which was not related to aging and which could occur either much later or much earlier than actual senility. Their 5145 meta-analyses showed that preventing frailty could contribute to life quality affecting either physical, psychological, or social aspects of human lives in various communities [53]. What deserves particular attention is a Swedish study which found that both the perception of one’s health and the social network which an individual belongs to constitute crucial components for a frail elderly person’s life satisfaction [53]. Kojima et al., in their subsequent investigations, recommended undertaking interdisciplinary and multifaceted activity in the area of public health policy concerning care treatment provided for people with frailty syndrome [24].

### 4.3. Healthy Practices and Preventive Behavior Correlation

In the authors’ own research, the means of both health practices (3.73 ± 0.85) and preventive behaviors (3.84 ± 0.87) were of higher value than the values for dialysis-treated patients (3.47 ± 0.82 vs. 3.67 ± 0.80), for men after cardiac treatment (3.31 ± 0.94 vs. 3.63 ± 0.85), and for women undergoing menopause (3.43 ± 0.62 vs. 3.71 ± 0.74) in normalized groups [10]. As the research reviews indicate, the health condition can be perceived by an individual as the sum of one’s own healthy behaviors [10]. One of the examples refers to the scientifically proven impact of physical activity on the mental well-being of a person. Scottish scientists examined 20,000 people and discovered a relationship between physical activity and the mental condition of a person. It was estimated that even low-level physical activity such as a walk, performing household chores, or gardening significantly contribute to reducing stress and anxiety [4]. Other studies showed that people who prefer a healthy lifestyle, as compared to those who choose to live unhealthily, have a more optimistic attitude to life and have higher self-esteem [10]. Furthermore, Harris investigated the relationship between physical activity and mood in a pre/post-community-wide, gamification-based intervention. The results indicated that improvements in mental well-being were significantly higher for the least active before the intervention. Moreover, a strong, positive link between an increase in physical activity and mental well-being was observed [54]. Healthy behaviors, defined as health practices, also included sleep routine. Many researchers have found that exercises and physical activity can improve sleep quality. A study conducted in Brazil including a group of individuals over 60 who were provided with a program, based on the recommendations of the American College of Sports Medicine, promoting physical exercises for the elderly, may serve as an example. The authors showed that a regular practice of partially supervised home-based exercises significantly improves the quality of sleep and reduces daytime drowsiness of confined elderly adults and may be considered therapeutic, safe, and easy to be implemented [42]. Although every person possesses several personal resources, the types of the resources triggered when required, as well as an assessment of their usefulness, are left to the discretion of an individual and depend on both the situational and social context. The authors’ own research proves that positive orientation constitutes a predictor of healthy practices, although it does not reflect the predictive power of preventive behaviors. This can raise some doubt when one considers taking care of one’s own health. Undertaking multifaceted and interdisciplinary actions, with the aim to modify healthy behaviors of both an individual as well as bigger communities, constitutes a global challenge due to the fact that chronic illnesses, according to World Health Organization (WHO), are likely to become a major factor contributing to disability by 2020 and are likely to generate the highest costs of dealing with health-related problems [55].

### 4.4. Implications and Limitations

The development of personal resources constitutes one of many elements crucial for coping with difficult life situations, one of which is represented by a chronic illness. When patients are able to actively approach the difficulties they face and can cope with emotions as well as the requirements imposed by the illness, their mental attitude improves. The study confirms that implementing modifications to healthy behaviors requires taking into consideration different determinants of behavior and health. What seems worth considering is the concept of enriching the rehabilitation program of chronically ill patients with workshops, conducted by a psychotherapist, devoted to the development of individual psychological resources, including positive orientation. The present study, however, contains some limitations connected with the cross-sectional character of the research and the retrospective. Another problem results from the small number of respondents as well as from the fact that the research also included patients who were subjected to rehabilitation resulting from stroke and who suffered from general motor control disorders which required specific therapeutic actions. As far as further studies are concerned, it is recommended to select a larger sample of patients with regard to their specific health problems and specific medical care requirements, to formulate precise recommendations pertaining to the development of particular mental resources, including positive orientation.

## 5. Conclusions

A significant majority of the research participants obtained average or high results in general health behaviors and positive orientation. As for the prediction of health-promoting behaviors, both positive orientation as well as age showed a varied degree of predictive power. No significant differences in the severity of overall gender-related health behaviors were observed.

As far as the severity of eating habits is concerned, positive orientation showed, similar to age, a level of predictive power and could positively affect the trend in modification of eating behaviors of the patients subjected to rehabilitation.

Positive orientation plays a significant role in the prediction of the severity of healthy behaviors related to mental positive attitude, which contributes to developing a more positive approach to one’s illness.

Positive orientation did not reveal the predictive power of the preventive behaviors, whereas age demonstrated a significant contribution. Compliance with health recommendations increased with the age of the respondents.

A lower contribution of positive orientation, compared to age, to the prediction of severity of healthy practices (which could either promote health or increase the risk of developing illnesses) was discovered to be apparent.

## Figures and Tables

**Figure 1 ijerph-16-03408-f001:**
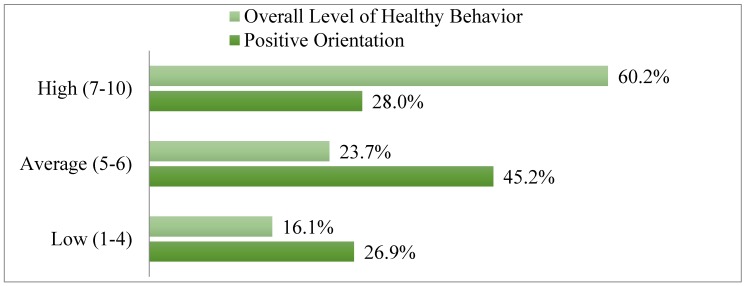
Comparison of data according to Sten scores—healthy behaviors and positive orientation.

**Figure 2 ijerph-16-03408-f002:**
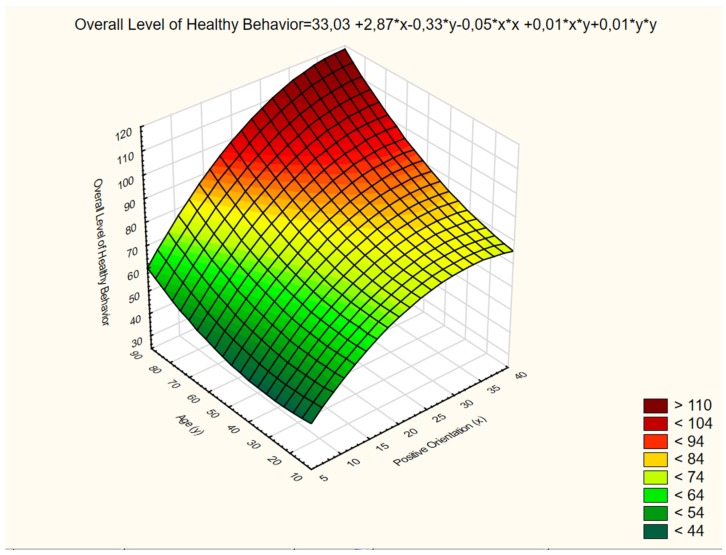
The predictors of healthy behaviors—correlation curve.

**Table 1 ijerph-16-03408-t001:** Positive Orientation Scale and Healthy Behavior Inventory—descriptive statistics.

Variables		M	SD	Me	Confidence −95%	Confidence +95%	Min.–Max.	Q1–Q3
Positive orientation		29.14	5.11	30	28.09	30.19	8–39	27–32
Healthy behaviors-general		91.39	17.56	95	87.77	95.00	35–120	79–104
Behavior categories	Proper eating habits	3.67	0.91	3.83	3.48	3.85	1.5–5	3–4.5
	Prophylactic behaviors	3.84	0.87	4.00	3.66	4.02	1.3–5	3.3–4.5
	Positive mental attitude	3.99	0.75	4.17	3.83	4.14	1.3–5	35–4.5
	Healthy practices	3.73	0.85	3.83	3.56	3.91	1.5–5	3.2–4.5

Explanation: M–mean, SD–standard deviation, Me–median, Min.–minimum, Max.–maximum, Q1–bottom quartile, Q3–upper quartile.

**Table 2 ijerph-16-03408-t002:** R-Pearson’s correlation coefficient of positive orientation with healthy behaviors, gender of respondents.

Variables		Number of Respondents n = 93	Gender	
			Women n = 46	Men n = 47
Health behaviors-general		0.29 **	0.21	0.30
Behavior categories	Proper eating habits	0.24 *	0.23	0.24
	Prophylactic behaviors	0.13	0.13	0.13
	Positive mental attitude	0.41 ***	0.34 *	0.45 **
	Healthy practices	0.25 *	0.20	0.27

Statistically significant: * *p* < 0.05; ** *p* < 0.01; *** *p* < 0.001.

**Table 3 ijerph-16-03408-t003:** Predictors of health behaviors—regression model.

Variables		R^2^	β–Standardized	β	Error–β	t	*p*-Value
Overall level of healthy behavior	Age	0.13	0.37	0.46	0.12	4.01	0.001
	Positive orientation	0.22	0.31	1.05	0.32	3.30	0.001
	Constant value			33.4	11.9	2.81	0.01
	R = 0.47; R^2^ = 0.22; corrected R^2^ = 0.21						
Proper eating habits	Age	0.06	0.25	0.02	0.01	2.57	0.01
	Positive orientation	0.12	0.25	0.04	0.02	2.54	0.01
	Constant value			1.41	0.65	2.16	0.03
	R = 0.34; R^2^ = 0.12; corrected R^2^ = 0.10						
Prophylactic behavior	Age	0.16	0.40	0.02	0.01	4.21	0.001
	Constant value			1.62	0.61	2.68	0.01
	R = 0.42; R^2^ = 0.16; corrected R^2^ = 0.16						
Positive mental attitude	Positive orientation	0.17	0.42	0.06	0.01	4.51	0.002
	Age	0.22	0.24	0.01	0.00	2.54	0.01
	Constant value			1.46	0.51	2.89	0.005
	R = 0.47; R^2^ = 0.22; corrected R^2^ = 0.20						
Healthy practices	Age	0.14	0.39	0.02	0.01	4.19	0.001
	Positive orientation	0.21	0.26	0.04	0.02	2.81	0.01
	Constant value			1.07	0.58	1.84	0.07
	R = 0.46; R^2^ = 0.21; corrected R^2^ = 0.19						

Statistically significant: *p* < 0.05; *p* < 0.01; *p* < 0.001. R–correlation coefficient, R^2^–multiple determination coefficient, β–Standardized regression coefficient, β–nonstandardized regression coefficient, Error β–nonstandardized regression coefficient error, t–t test value.

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
