# Peer review of "Positive Orientation as a Predictor of Health Behavior during Chronic Diseases"

_ijerph, 2019, doi:10.3390/ijerph16183408_

Round 1
Reviewer 1 Report
I do appreciate the efforts of the authors to improve the quality of the manuscript.
The modifications made to the statistical section make sense now.
Also, specifying the patients` disabilities, especially the neurological ones, make the readers better understand the manuscript.
I also appreciated the fact that authors confronted their results on the basis of the concept of frailty, that really fits in your discussion, based on your results. I insist on this part because in the geriatric, neurological and rehabilitation field considering the physical frailty is an important issue, especially for old and hospitalized people (that usually overlap) in a broad variety of diseases, considering also about of co-morbidities (Fried, et al. 2001; Koch et al., 2013). This is why I was suggesting in my first review that a more detailed description of the patients enrolled in the study (co-morbidity, degree of impairment, cognitive demise) helps us in understanding the burden of the disease, and in identifying the factors that help us in understanding and hopefully predicting the the course of the disease (and of the rehab program) in order to undertake the actions needed to improve the outcome of the rehab program (Koch et al., 2013). As clinicians working in these departments we have to have this aspect well built-up in our minds.
Suggested literature
-Koch G, Belli L, Giudice T Lo, Lorenzo F Di, Sancesario GM, Sorge R, Bernardini S, Martorana A (2013) Frailty among Alzheimer’s disease patients. CNS Neurol Disord Drug Targets 12, 507-511.
-Fried LP, Tangen CM, Walston J, Newman AB, Hirsch C, Gottdiener J, Seeman T, Tracy R, Kop WJ, Burke G, McBurnie MA; Cardiovascular Health Study Collaborative Research Group. Frailty in older adults: evidence for a phenotype. J Gerontol A Biol Sci Med Sci. 2001 Mar;56(3):M146-56.
Author Response
The authors of the study would like to thank you very much for the thorough substantive editorial manuscript assessment and positive feedback. We would like to inform you that we have adapted to the reviewer's suggestions and introduced two literature items. In addition, we would like to inform you that in the list of references we have removed item "12" because it was a repetition of item "3". The changes made to the manuscript are highlighted in red.

Reviewer 2 Report
I highly appreciate the authors’ sincere responses to my previous comments. In this revision, it was much easier to understand the statistical methods and the explanation of the contents. I think the paper is in better shape and met the conditions to be accepted for publication.
Author Response
The authors of the study would like to thank you very much for the thorough substantive editorial manuscript assessment and positive feedback.
This manuscript is a resubmission of an earlier submission. The following is a list of the peer review reports and author responses from that submission.
Round 1
Reviewer 1 Report
The aim of this study was to determine whether the positive orientation is an important predictor of health behaviors of patients with neurological dysfunction and with chronic movement disorders which require a rehabilitation program. In general, the paper is well written and studies an important topic. However, the introduction and discussion are not persuasive enough that the findings make a significant contribution to the literature and could, therefore, override these limitations. I include some comments below related to this summary for consideration.
Introduction:
1. In relation to the contribution of the study to the literature, I did not get a sense from the article that the findings revealed anything other than what we already know.
2. The introduction of the paper was very descriptive, it did not situate the current study in literature or highlight what the gap in the literature is that this study is trying to address.
3. The references used are too old. Authors must make an effort, in order to use more recently references about this topic;
4. Another concern is related to the literature gap. It is unclear what the gap that you intend to fill is?
5. The hypothesis justification is very poor. Why do you believe in this hypothesis?
6. Finally and more importantly, what is the theoretical framework, which supports the present study?
7. Overall, the introduction needs to be reworded, considering my previous comments
Methodology
The methods section lacks:
1- the recruitment date range (month and year);
2- a description of any inclusion/exclusion criteria that were applied to participant recruitment;
3- a statement as to whether your sample can be considered representative of a larger population,
5- a description of how participants were recruited, and descriptions of where participants were recruited and where the research took place.
6- Regarding measure, I suggest that in addition to the internal consistency values, the adjustment values of the measurement model should also be reported.
Statistical Analysis
1) Please present the results from normality tests as well as, which normality test was used;
2) Please, report in detail each step regarding multi-regression analysis;
3) Please, include the bivariate correlation magnitude interval
Results
Results are well reported
Discussion:
I will not provide detailed comments on the discussion as I believe it is likely to change quite a bit if you follow my previous recommendations. But I will provide some general comments.
I also suggest that you order the discussion according to the study aims and use sub-headings for each part of the discussion.-
You present a lack of limitations of the present study and in my opinion the paper have much more, which you should consider.
Overall, the discussion is very descriptive and any statements about the contribution and conclusions of the study are not new. What is the contribution to the literature, what is interesting about your results? The issues with the introduction about the lack of appropriate and too old literature on this topic are replicated in the discussion, as well as, the lack of connection between empirical and theoretical evidence.
In sum, I think the article needs a number of considerable revisions in order to be considered for publication in the International Journal of Environmental Research and Public Health.
Reviewer 2 Report
In this paper, Kupewicz and co-workers aimed at investigating whether positive orientation is an predictor of health behaviors of patients with neurological dysfunction and with chronic movement disorders involved in a rehabilitation program. By using a Positive Orientation Scale and a Health Behavior Inventory authors were able to measure the intensity of health-promoting behaviors. In 94 patients undergoing rehabilitation program they found out that age and positive orientation were the best predictors of overall healthy behavior.
The results of this research might be important in order to better tailor rehabilitation programs in chronic patients; however there are several concerns that need to be addressed before publication.
1) The whole manuscript needs a deep and thorough revision for English language. Several mistakes can be depicted all along the whole manuscript.
2) I appreciate the fact that authors introduced the topic in a very detailed way, but, in my personal opinion, for the sake of the readability of the manuscript and the interpretation of the results of the research, I would redistribute or facing deeply some topics in the discussion in light also of the results.
3) Authors must specify which kind of “motoric disorder” patients are suffering: is this a neurological problem, orthopedic, systemic? Which kind of physical limitation are they actually suffering? (hemiparesis, gait disturbances, leg adduction/leg abduction). All these info are important in order to better understand who can be the target of a tailored rehab program. Please provide these information.
4) 5 patients suffering from neurological disorder is not an acceptable group, especially if compared to the “motoric syndrome” group. As you are not running analyses to differentiate the two groups, it would be worthy to consider patients as a unique group suffering from different problem that need a specific rehab program.
5) Disease duration is one of the info that need to be taken in account for the analyses. I invite authors to add this, if they own the data.
6) Again, the specification of the patients suffering from specific disease and consequent physical limitation is very important, according to my personal view. Indeed, specifying which neurological dysfunction they are suffering is important in order to understand also the natural history of the disease. For example we know that the natural progression of Stroke patients (according to the burden of their lesions and to other pathological and personal factors such as comorbidity, collateral flow, etc)is to recover a certain quote of disability, while for example neurodegenerative disease patients (in primis Alzheimer`s disease patients but also other neurodegenerative disease such as Sopranuclear Progressive Palsy or MultySistem Atrophy) tend always to progress without significant improvements of their symptoms. Different is instead the natural evolution of Parkinson Disease patients, in which, hopefully, pharmacological approach is able to impact, especially in the first phases of the disease, the disease progression and especially quality of life.
7) In my opinion, especially looking at the outcome of your research, it would be interesting to introduce in the discussion also the concept of Frailty (Kojima et al., 2019), both social (Bunt et al., 2017) and physical (Fried et al. 2001), because this aspect is important part of the interpretation of your results. Again, a better categorization of your patients (neurological and non-neurological), the presence of comorbidity, the degree of their impairment (cognitive or motor) are all important factors for understanding the pathological burden of the disease (Koch et al., 2013) and, especially, which are the factors that can help us in understanding and hopefully modify the disease course in order to better understand which are the actions that can be undertaken to improve the outcome of the specific rehab program. (Koch et al, 2013)
8) This reviewer is unconvinced about the arbitrary cutoff used for the age of the participants to this study (results in Table 4). How many were the subjects who were right on one side or the other of the cutoff? If so, then it might be more prudent to assess all patients as a single group with age as a covariate, as done in the later analyses. Separating patients into two subgroups does not appear to be necessary to test the hypotheses of interest. The authors could simply combine older and younger participants and test for differences in a multiple linear regression analysis and include as covariates, the variable age, disease duration, and their interactions with the positive orientation and overall healthy behavior.
9) The manuscript would benefit, from an outlook point of view, of a graphical editing. I suggest to show the curve of correlation between age and positive orientation as precidtors of overall healthy behaviors.
Suggested literature:
Kojima G, Liljas AEM, Iliffe S. Frailty syndrome: implications and challenges for health care policy. Risk Manag Healthc Policy. 2019 Feb 14;12:23-30. doi: 10.2147/RMHP.S168750. eCollection 2019. Review.
Bunt S, Steverink N, Olthof J, van der Schans CP, Hobbelen JSM. Social frailty in older adults: a scoping review. Eur J Ageing. 2017 Jan 31;14(3):323-334. doi: 10.1007/s10433-017-0414-7. eCollection 2017 Sep. Review.
Fried LP, Tangen CM, Walston J, Newman AB, Hirsch C, Gottdiener J, Seeman T, Tracy R, Kop WJ, Burke G, McBurnie MA; Cardiovascular Health Study Collaborative Research Group. Frailty in older adults: evidence for a phenotype. J Gerontol A Biol Sci Med Sci. 2001 Mar;56(3):M146-56.
Koch G, Belli L, Giudice T Lo, Lorenzo F Di, Sancesario GM, Sorge R, Bernardini S, Martorana A (2013) Frailty among Alzheimer’s disease patients. CNS Neurol Disord Drug Targets 12, 507-511.